

# An intercomparison of stratospheric gravity wave potential energy densities from METOP GPS-radio occultation measurements and ECMWF model data

Markus Rapp[1,2], Andreas Dörnbrack[1], and Bernd Kaifler[1]

[1]Deutsches Zentrum für Luft- und Raumfahrt, Institut für Physik der Atmosphäre, Oberpfaffenhofen, Germany
[2]Meteorologisches Institut München, Ludwig-Maximilians-Universität München, Munich, Germany

*Correspondence to:* Markus Rapp (markus.rapp@dlr.de)

**Abstract.** Temperature profiles based on radio occultation (RO) measurements with the operational European METOP-satellites are used to derive monthly mean global distributions of stratospheric (20 - 40 km) gravity wave (GW) potential energy densities ($E_P$) for the period July 2014 - December 2016. In order to test whether the sampling and data quality of this data set is sufficient for scientific analysis we investigate to which degree the METOP-observations agree quantitatively with ECMWF

operational analysis (IFS-data) and reanalysis (ERA-Interim) data. A systematic comparison between corresponding monthly mean temperature fields determined for a latitude-longitude-altitude grid of 5° by 10° by 1 km is carried out. This yields very low systematic differences between RO and model data below 30 km (i.e., median temperature differences is between -0,2 and +0,3 K) which increases with height to yield median differences of +1,0 K at 34 km and +2,2 K at 40 km. Comparing $E_P$-values for three selected locations at which also ground based lidar measurements are available yields excellent agreement

between RO and IFS-data below 35 km. ERA-Interim underestimates $E_P$ under conditions of strong local mountain wave forcing over Norther Scandinavia which is apparently not resolved by the model. Above 35 km, RO-values are consistently much larger than model values which is likely caused by the model sponge layer which damps small scale fluctuations above ∼32 km altitude. The comparison between RO and lidar data reveals very good qualitative agreement in terms of the seasonal variation of $E_P$, however, RO-values are consistently smaller than lidar values by about a factor of two. This discrepancy

is likely caused by the very different sampling characteristics of RO and lidar observations. Direct comparison of the global data set of RO and model $E_P$-fields shows large correlation coefficients (0.4 - 1.0) with a general degradation with increasing altitude. Concerning absolute differences between observed and modelled $E_P$-values, the median difference is relatively small at all altitudes (but increasing with altitude) with an exception between 20 and 25 km where the median difference between RO- and model-data is increased and where also the corresponding variability is found to be very large. The reason for this is

identified as an artifact of the $E_P$-algorithm: this erroneously interprets the pronounced climatological feature of the tropical tropopause inversion layer (TTIL) as GW activity hence yielding very large $E_P$-values in this area and also large differences between model and observations. This is because the RO-data show a more pronounced TTIL than IFS and ERA-Interim. We suggest a correction for this effect based on an estimate of this 'artificial' $E_P$ using monthly mean zonal mean temperature profiles. This correction may be used in the future to study, for example, the annual cycle of zonal mean GW activity using the

here considered data.



# 1 Introduction

It has long been known that momentum and energy transport by gravity waves (henceforth abbreviated as GW) is of major importance for the mean thermal and dynamical state of the middle atmosphere (Lindzen, 1981; Holton and Alexander, 2000). Being mainly excited in the troposphere by flow over terrain, by convection, or by spontaneous emission, GW may propagate

both vertically and horizontally over large distances to deposit their momentum and energy upon instability or transience far away from their source (e.g., Fritts and Alexander, 2003; Sato et al., 2009; Preusse et al., 2009; Sato et al., 2012; Bölöni et al., 2016). Thus, GW are a major means to couple the middle and upper atmosphere to the troposphere (e.g., Lübken et al., 2010, and references therein). In addition, it has recently been shown that GW also couple the middle atmosphere downward to the troposphere (Kidston et al., 2015, and references therein). With minimum horizontal scales as small as 10 km GW must

still be parameterized in global climate models with typical horizontal resolutions of a few hundred kilometers. Hence, the development of physics-based parameterizations of GW and their effect on the mean flow have been identified as a major research focus in the climate research community (Shepherd, 2014).

Given this large importance of GW, it is not surprising that efforts have been undertaken and are under way trying to characterize GW sources, their propagation, as well as their dissipation and wave-mean flow interaction with complementary

experimental, theoretical and numerical techniques (see, e.g., Fritts and Alexander, 2003; Plougonven and Zhang, 2014; Fritts et al., 2016; Wagner et al., 2017; Sutherland, 2010; Nappo, 2012, for recent reviews, overview papers, and text books). Ground based remote sensing with lidars and radars and in-situ observations with balloons, research aircraft, and sounding rockets are critically important for process studies. However, global satellite observations are needed to determine dominant tropospheric source regions and processes as well as global propagation pathways and the resulting gravity wave drag imposed on the

mean flow to constrain GW parameterizations for climate and weather prediction models (Alexander et al., 2010; Geller et al., 2013). Since the pioneering work by Fetzer and Gille (1994),Wu and Waters (1996), and Eckermann and Preusse (1999) there have been many attempts to characterize the global distribution of gravity wave activity using such different remote-sensing techniques as Limb (e.g., Ern et al., 2004; Preusse et al., 2009; Ern et al., 2011; Zhang et al., 2012) and Nadir sounders (e.g., Hoffmann et al., 2016; Ern et al., 2016), as well as GPS-based radio occultation measurements (e.g., Tsuda et al., 2000; Hei

et al., 2008; Schmidt et al., 2008; Fröhlich et al., 2007; Hindley et al., 2015; Šácha et al., 2015; Khaykin et al., 2015; Khaykin, 2016; Schmidt et al., 2016).

This paper focusses on the derivation of gravity wave potential energy densities ($E_P$) from GPS radio occultation (RO) measurements onboard the operational METOP-A and METOP-B-satellites operated by EUMETSAT (=European Organisation for the Exploitation of Meteorological Satellites) and the subsequent systematic comparison of $E_P$-fields with ECMWF

(European Center for Medium range Weather Forecast) operational forecast and reanalysis data. This is done to answer the question whether the sampling and data quality of the two operational METOP-satellites is sufficient to characterize the global stratospheric gravity wave activity (measured in terms of $E_P$) on a monthly basis. Furthermore, we investigate whether the METOP-observations agree quantitatively with the ECMWF model-fields such that the latter can be used for the interpretation of observational results. Accordingly, this paper is organized as follows: in Section 2 we describe the data base of METOP-A



and METOP-B radio occultation temperature data obtained between July 2014 and December 2016. In addition, we give a brief introduction to the ECMWF data sets used for comparison with the RO-data. We compare both temperature data sets (RO and ECMWF-data) as a baseline for the subsequent comparison of derived $E_P$-values. In Section 3 we describe our approach to derive $E_P$ followed by Section 4 where we thoroughly compare RO $E_P$-data to corresponding ECMWF-data sets. Similarities

and differences are discussed in Section 5 in which we will also derive a correction for erroneous interpretation of the tropical tropopause inversion layer (TTIL) as gravity wave activity. Finally, the major findings of this study are summarized in Section 6 in which also suggestions for future work will be made.

## 2 Data base

### 2.1 METOP-A/B GPS radio occultation data

The METOP-A and -B satellites orbit the Earth in a polar low Earth orbit and are the platforms for a variety of instruments supporting the European Weather Services including the Global Navigation Satellite System Receiver for Atmospheric Sounding (GRAS) with which GPS radio occultation measurements are performed delivering tropospheric humidity and tropospheric and stratospheric temperature profiles. During typical months, these two satellites record a total of $\sim$ 35.000 - 40.000 radio occultations. A typical sampling pattern in terms of the latitude and longitude distribution of the number of RO per month

is shown in Figure 1. This sampling is determined by the orbital geometry of the METOP-satellites on the one hand and the GPS-satellites on the other. Figure 1 reveals that there are typically between 10 and 50 occultations per 5 degree latitude and 10 degree longitude interval with maximum sampling at latitudes between 20 - 60° North and South and minima near the poles and at the equator. Note that we will use a corresponding gridding (i.e., $36 \times 36$ 5° latitude $\times$ 10° longitude bins) throughout this entire study.

The METOP RO-data are provided by the Radio Occultation Meteorology Satellite Application Facility (ROM SAF) on an operational basis in near real time and can be downloaded from www.romsaf.org. The primary measured quantity is the bending angle of the GPS radio waves as they transverse the refracting atmosphere. From bending angle profiles corresponding refractivity profiles can be derived from which in turn also temperature profiles can be determined (Kursinski et al., 1997). The latter can be done by either assuming that the refraction is entirely due to dry air (resulting in so-called 'dry' temperatures) or

by accounting for tropospheric water vapor by using additional information, e.g., from operational numerical weather forecast data in the framework of a one dimensional variational algorithm that uses ECMWF IFS data as a priori information (ROM SAF, 2014a, b, and references therein). The latter approach is pursued by the ROM SAF and corresponding temperature data are denoted 'wet' temperatures. For the current study we will mainly use dry instead of wet temperatures since the latter have been derived using model output and might hence not be considered as 'pure' measurements. Nevertheless, we will also briefly

consider wet temperatures and compare them to the more 'original' dry ones. Note that the ROM SAF provides dry and wet temperatures from July 2014 onwards only. Hence, in this study we restrict ourselves to the period from July 2014 to December 2016, i.e., a total of 30 months of data.



METOP temperature profiles are provided on geopotential heights which will be used here as the vertical coordinate. The fundamental vertical resolution of the technique, $\Delta z$, is limited by diffraction as the GPS rays pass through the atmosphere and results in about $\Delta z$=1 - 1.4 km in the altitude range between 15 and 40 km. Over this vertical interval, the horizontal line-of-sight resolution can be estimated to be around 190 - 270 km due to the limb geometry of the observations (see Kursinski et al.,

1997; Hindley et al., 2015, for details).

## 2.2  ECMWF operational analysis and reanalysis data

For comparison to the METOP RO-data we use two different data sets provided by the ECMWF: one is the operational analyses from the Integrated Forecast System (IFS). These have a horizontal resolution of about 8 km (T1279) and were evaluated on 25 pressure levels between 1000 and 1 hPa which we converted to geopotential heights and interpolated them on a regular vertical

grid with 1 km spacing. Model output is available every six hours. Details about the model can be found in Malardel and Wedi (2016) and in references therein.

The second model data set that we use is the ERA-Interim reanalysis. ERA-Interim is a global atmospheric reanalysis from 1979 which is based on a 2006 release of the IFS. The spatial resolution of the data set is approximately 80 km (T255). For the current study, model fields were evaluated on 37 standard pressure levels between 1000 and 1hPa which we converted to

geopotential heights and interpolated them on a regular vertical grid with 1 km spacing. For details about ERA-Interim see Dee et al. (2011).

Please note that ECMWF does assimilate RO-bending angle data (among many other data sets) from a variety of instruments including (but not limited to) the METOP-data for both the ERA Interim reanalysis as well as for the IFS analyses (see Poli et al., 2010, as well as the ECMWF web site).

## 2.3  Comparison between RO and ECMWF-temperature data

In this subsection we systematically compare RO temperatures with ERA-Interim and IFS model data. As a start, Figure 2 shows zonal mean temperatures for the months March, June, and December 2015 derived from METOP GPS RO-dry data (left column), GPS-RO wet data (middle column), and from ERA Interim. Note that from now on, we will refer to METOP GPS-RO-dry and -wet data as 'RO-dry' and 'RO-wet' data for brevity. Overall, all data sets agree well with notable differences,

however, between the dry temperatures and the other two data sets in the troposphere and at the highest altitudes above 40 km. These findings are not surprising given that the retrieval for wet temperatures uses ECMWF IFS data as a priori information, that the assumption of dry conditions is certainly violated in the troposphere, and that the quality of RO observations in general decreases significantly above $\sim 40$ km altitude (Marquardt and Healy, 2005). In the following, we hence restrict our comparison to altitudes between 20 - 40 km.

For a more quantitative comparison, we have binned the ECMWF data sets on the same space and time grid as the RO-data, i.e., mean profiles were determined for the period of one month and a latitude-longitude-altitude grid of 5° by 10° by 1 km. Figure 3 shows corresponding scatter plots between RO dry temperatures and corresponding IFS-data for all 30 months of data considered in this study (i.e., July 2014 - December 2016) as well as histograms of the temperature differences between





the data for three selected altitudes. This reveals very large correlation coefficients close to one between the data with a general degradation of the (still very good) correlation as well as an increasing bias between the data with increasing altitude. The full altitude variation of the correlation coefficients between the considered data sets as well as the median absolute temperature differences along with corresponding 10% and 90% percentiles is shown in Figure 4. This again shows an almost

perfect correlation between ERA-Interim and IFS-data (as expected) as well as between the RO-wet temperatures and the IFS. Again, only the dry temperatures show a notable disagreement from the other data sets at altitudes above $\sim 35$ km. This is further quantified with the median biases (and percentiles) shown in the right panel of Figure 4 which shows a median bias of +1 K (+2 K) between RO-dry temperatures and the IFS (i.e., IFS temperatures are larger than dry RO temperatures) at 34 km (40 km) km with a corresponding large variability range as indicated by the percentiles. We note that these values are

in excellent quantitative agreement with a previous study in which GPS RO-observations were compared to ECMWF-data (Scherllin-Pirscher et al., 2011). Compared to this bias of the RO-dry data, it is again not surprising to see that the RO-wet temperatures show a much smaller bias (close to zero) to the IFS and that also the corresponding uncertainty range is much reduced. Both derived biases and uncertainty ranges agree well with previous findings of an analysis of the ROM SAF as described in the corresponding validation report (ROM SAF, 2014c). In all, RO-temperatures agree well with ERA-Interim

and IFS-temperatures such that it appears justified to proceed and next compare corresponding $E_P$-values.

## 3 Derivation of $E_P$

We next turn to the derivation of $E_P$ from the various input temperature data sets considered in this study. $E_P$ is defined as follows:

$$E_p(z) = \frac{1}{2} \frac{g^2}{N^2(z)} \overline{\left( \frac{T'(z)}{T_0(z)} \right)^2} \tag{1}$$

where $g$ is acceleration of gravity, $N$ is the bouyancy-frequency, $T'$ is the temperature perturbation owing to the GW, and $T_0$ is the background temperature. The overbar denotes averaging, which is here carried out over the spatial domain of the latitude-longitude-grid and the time period of one month. In Equation 1 all quantities depend on height $z$ except for $g$ for which we use a constant value of 9.81 ms$^{-2}$. The main challenge in deriving $E_p(z)$ from measured temperature profiles lies in the separation between background and perturbations. Different studies have used various approaches such as filtering of profiles in the

vertical or in the horizontal provided that the horizontal sampling is sufficient. See Khaykin (2016) and Ehard et al. (2015) for recent critical discussions of the advantages and disadvantages of different techniques. For this study, we follow the approach of Ehard et al. (2015), i.e., we apply a fifth-order Butterworth filter with a cutoff wavelength of 15 km to vertical temperature profiles from both the RO-data as well as from ERA Interim and the IFS. Resulting $T_0(z)$-profiles are then used to derive corresponding $N^2(z)$-profiles from $N^2 = \frac{g}{T_0} \cdot (\frac{dT_0}{dz} + \frac{g}{c_p})$ where $c_p$ is the specific heat capacity of air for constant pressure.

Corresponding arbitrarily chosen sample profiles from RO-dry data are shown in Figure 5. Figure 5 shows both cases with strong (middle panel) and weak GW-activity (left panel). These sample profiles further show that the background temperature determination has weaknesses in cases with a very pronounced tropopause as in the right panel. We will come back to this





issue in more detail in Section 5. Here, neither the pronounced tropopause (at around 17 km), nor the inversion layer above (i.e., between 20 - 25 km) is well captured by the Butterworth filter resulting in unrealistically large temperature perturbations which might not be confused with real gravity wave-induced temperature perturbations. This is a general problem with all techniques analyzing vertical temperature profiles which has motivated many authors to exclude the tropopause region and the

lowest altitudes above it from further analyses (see e.g. Schmidt et al., 2008, for a detailed discussion and an approach to derive GW properties in the vicinity of the tropopause). For this reason, we will exclude altitudes below 20 km from our analysis and focus on the altitude range between 20 and 40 km only, knowing, of course, that the largest altitudes need to be treated with care since noise of RO-data is known to pick up significantly above ∼35 km altitude (Marquardt and Healy, 2005).

## 4  Comparison of METOP $E_P$-values with ECMWF model data and ground based lidar measurements

We next present a systematic comparison of $E_P$-values derived from METOP RO-dry temperatures, from the IFS, and from ERA-Interim. As an initial impression, Figure 6 shows monthly mean latitude-longitude cross sections of $E_p$ at selected altitudes of 30, 33, 36, and 39 km for December 2015, respectively. At 30 km, the RO-data reveal pronounced GW activity over Scandinavia, over the Iberian peninsula and North Africa, and in a band in the vicinity of the equator with strongest activity in the tropical central Pacific (135 - 180° E). Moving to 33 km altitude, $E_P$-values increase with pronounced activity still over

Scandinavia, strong activity at around 40°N in the Atlantic storm track region, and an additional activity center over the northern part of South America. At larger altitudes, these general features remain, but become smeared out geographically. Generally speaking, this overall morphology of GW activity is well reproduced by both the IFS and ERA-Interim with some notable differences. First of all, $E_p$-values from the IFS and ERA-Interim are generally smaller than corresponding RO-values, with this discrepancy increasing with increasing altitude. Secondly, ERA-Interim does not capture the GW activity over Scandinavia

that is clearly seen in the RO-data and also in the IFS-data. The latter finding is likely due to the significantly coarser horizontal resolution (and hence also a coarser resolution of orography) which keeps the ERA-Interim reanalysis from capturing rather localized orographic gravity wave activity as the one here seen over Scandinavia.

For a more detailed comparison, we have next extracted time series of $E_P$ for different altitude bands at selected locations. We have selected three locations at which we have conducted extended ground based Rayleigh lidar observations of $E_P$ and

are hence in a position not only to compare RO-data with the two different model data sets but also with the ground based data. These locations are: Lauder, New Zealand (45° S, 169.7° E), where ground based Rayleigh lidar measurements were conducted from June through December 2014 (Kaifler et al., 2015; Fritts et al., 2016), Sodankylä, Finnland (67° N, 26° E), where observations were taken from September 2015 until May 2016 (Kaifler et al., 2017), and finally in the German Bavarian Forest (48.8° N, 13.7° E) with measurements from May until December 2016. Figure 7 shows time series of monthly mean

$E_P$ from July 2014 through December 2016 for these locations and for the altitude ranges 15-25 km, 25-35 km, and 35-45 km, respectively. Note that we have binned the model data to the same latitude-longitude grid as the RO-data for a proper comparison. Figure 7 overall reveals a very good fit between RO and model data: in the altitude range from 15-25 km, RO and model data fit very well both in terms of absolute values as well as in terms of month-to-month variation for all three





locations (top panels). At altitudes between 25-35 km (middle panels), the agreement is still very good, however, peak RO values are underestimated by both models. This is particularly pronounced for Sodankylä (left), where local mountain wave activity is likely causing the strong wintertime $E_P$-peak. Consistent with the results shown in Figure 6 this peak is qualitatively well reproduced (but still slightly underestimated) by the IFS but completely missed by ERA-Interim due to the much coarser

horizontal resolution of the latter. Finally, at the highest altitudes, the overall seasonal variation of $E_P$ that is observed with the RO-sensors is reproduced by the models, but modelled $E_P$-values are smaller than those derived from RO-observations by factors between 2 and 3. This is as expected since the sponge layer in the ECMWF-models starts strongly damping any small scale structures above 10 hPa or ∼32 km (Jablonowski and Williamson, 2011; Ehard, 2017).

     Next, we compare the same RO time series to local $E_P$-observations obtained with Rayleigh-lidar (see Figure 8). The

portable lidar systems as well as the data analysis procedure used during the three campaigns have been described in detail in Kaifler et al. (2015) and Kaifler et al. (2017), respectively. In short, Rayleigh lidar measurements yield relative density profiles at altitudes where pure molecular scatter accounts for the signal, i.e., from above the stratospheric aerosol layer. Hence, data are available for altitudes above ∼30 km (and below ∼90 km) but may be extended to lower altitudes after careful analysis making sure that stratospheric aerosol-scatter did not contribute to the signal. Relative density profiles are then converted to

temperatures applying hydrostatic downward integration. Finally, $E_P$-values are derived in the same manner as for the RO-data described above (see Section 3).

     The comparison shown in Figure 8 reveals that lidar and RO-data generally show very similar seasonal variation. However, the comparison also shows that the local lidar observations yield significantly larger $E_P$-values by up to a factor of ∼2. This is likely because the lidar observations are sensitive to a larger part of the gravity wave spectrum than the RO-observations. As

described in Section 2.1, the horizontal line of sight of RO-observation is approximately 190 - 270 km. Hence, depending on the orientation of the wave vector relative to this line of sight, the RO-technique may not resolve waves with horizontal wavelengths shorter than these 190 - 270 km (if the phase fronts are aligned with the line of sight; the RO technique might, however, be able to detect GW with shorter horizontal wavelengths if the phase fronts are perpendicular to the line of sight, see Kursinski et al. (1997) and de la Torre and Alexander (2005) for details). Hence, it is clear that RO-observations are only sensitive to

GW with rather large horizontal wavelengths whereas lidar observations may also detect much smaller scale gravity waves. In addition, we also need to realize that the spatial sampling for both data sets is very different: while the $E_P$-values based on RO-data are typically based on 20 - 40 single (snapshot) temperature profiles that have been obtained in a geographical area of 5° in latitude and 10° in longitude, the lidar data shown here are based on 10-20 nightly means each consisting of several hours of GW observations (see lower panels in Figure 8). While it is difficult to assess the quantitative impact of this very different

sampling on the resulting $E_P$-values, it is conceivable that the large geographical area over which the RO-data are obtained might result in a smearing out of local GW maxima and should hence tend to smaller values compared to local observations.

     In all, we conclude from the comparison of time series at the three considered locations that the fit between GPS-RO and IFS and ERA-Interim data is generally very good whereas comparison to local observations indicates that RO-observations are low-biased - which is likely due to different observational filters of both techniques (see, e.g., Alexander et al., 2010; Ern et al.,

2004, for a thorough discussion of observational filters of different techniques). Next, we finally compare GPS-RO with IFS





and ERA-Interim data on a global basis. For all thirty months between July 2014 and December 2016 we have computed $E_P$ on a grid of 5° in latitude, 10° in longitude, and 1 km in the vertical for the whole considered altitude range of 20 - 40 km. For each altitude, we have then analyzed the relation between the two GPS-RO data sets and the model data sets in terms of correlation coefficients as well as in terms of absolute differences. An initial impression of the statistical relation between $E_P$-values from

RO-dry data and from IFS-data is presented in Figure 9 which shows corresponding scatter plots along with a linear regression to the data as well as histograms of the absolute difference between the two data sets for three selected altitudes. Figure 9 shows a very large correlation of R=0.94 at 22 km, a minimum value of R=0.45 at 28 km, and a slightly larger value of R=0.56 again at 38 km altitude. Furthermore, it is common to all three histograms that IFS-values are biased low with respect to the RO-data. Interestingly, though, the distribution is broadest at the lowest considered altitude with much narrower distributions above.

The complete altitude variation of correlations as well as biases is shown in Figure 10 which shows correlation coefficients as well as median differences (along with 10% and 90% percentiles) between ERA-Interim and IFS, between RO-dry data and IFS-data, and last not least between RO-wet data and IFS-data. Figure 10 shows several interesting features. Starting with the correlation coefficients, those are generally large (between 1,0 and 0.5) except for the altitude range between 25 km and 30 km where the correlation of both RO-data products (wet and dry) with model data show a minimum with values as low as 0.4.

Above 30 km, however, correlations coefficients increase again. Besides this striking minimum between 25 km and 30 km, the overall envelope of the altitude variation shows larger correlation coefficients between 0.9 and 1.0 below 25 km and values between 0.8 (for the correlation between ERA Interim and IFS) and 0.5 (for the correlations between the RO-dry data and IFS data) at 40 km. Turning to absolute differences (right panel in Figure 10), the median differences between ERA-Interim and IFS-data are very small (less than 1 J/kg) with IFS-values being slightly larger than ERA-Interim values. Concerning the

absolute differences between RO- and IFS-data, both RO data products yield systematically larger $E_P$-values than the IFS where, however, the median difference between the RO-wet data and the IFS-data is significantly smaller than the difference between the more 'original' RO-dry data and the IFS-data. Interestingly, both the median difference as well as its variability (indicated by the percentiles) is quite large at 20 km and decreases significantly up to an altitude of 25 km above which both median differences and related variability increases again up to the maximum altitudes considered.

**5  Discussion**

In order to identify the reason for the reduced correlation between RO and IFS-data between 25 km and 30 km as well as the relatively large bias below 25 km, we next consider a comparison of latitude-longitude distributions of $E_P$-values at selected altitudes based on RO-dry data and IFS-data. Corresponding results for December 2015 and June 2015 are presented in Figures 11 and 12, respectively. We start with a discussion of the relatively low correlation coefficients at altitudes between

25 and 30 km. Inspection of Figures 11 and 12 reveals that the likely reason for this is that apparently the IFS is hardly simulating any gravity wave activity at these altitudes whereas the observations do show some weak but clearly detectable GW activity. The reason why the IFS does not simulate any (respectively very weak) GW activity in the considered vertical wavelength range at these altitudes is not cleat at this point but is consistent for all months considered in this study and should



be further investigated in the future. As for the bias at altitudes below 25 km, the $E_P$-distributions shown at 20 and 22 km show that the strongest (apparent) GW activity is here observed in a band of $\pm 20°$ around the equator with significantly larger values seen in RO-data than in IFS-data. This is, however, the region of the tropical tropopause and its related TTIL. Note that it is on purpose that we refer to the tropical tropopause inversion layer as TTIL instead of the more commonly known

TIL, since the latter term has usually only been used for the mid latitude TIL and not the tropical one that we are dealing with here (Birner et al., 2002, 2006; Pilch Kedzierski et al., 2016). That this is indeed the case for the here considered data set is demonstrated in Figure 13 which shows zonal mean $N^2$-values based on RO and ERA-Interim data, respectively. Note that we have chosen to compute $N^2$-values from ERA-Interim and not IFS-data since our analysis in Figures 4 revealed very good agreement between ERA-Interim and IFS-temperatures. Figure 13 clearly shows that it is indeed the latitude and altitude range

of the TTIL which coincides with corresponding regions of large $E_P$-values in the considered data sets. In addition, Figure 13 also shows that the TTIL is more pronounced in the RO-data than in the ERA-Interim data. Hence, it is tempting to speculate that the large $E_P$-values seen in the tropics and the corresponding large differences between the RO-data and the IFS-data is because our algorithm to derive $E_P$-values from temperature profiles by means of separating background temperatures from gravity wave induced disturbances fails in this altitude and latitude region. In order to test this idea further, we present zonal

mean $E_P$-values as a function of latitude and altitude between 20 and 40 km altitude based on both RO-dry-data and ERA-Interim-data in Figure 14. This Figure clearly shows the region of large $E_P$-values between 20 and 25 km altitude and at latitudes between -20° to +20°. It also shows that RO-values in this region are significantly larger than in the ERA-data set. In order to test whether these are indeed real indications of gravity wave activity or rather artifacts due to the TTIL we have next applied our algorithm to derive $E_P$-values to monthly mean zonal mean temperature profiles. For those, it can safely be

assumed that they do not contain any remaining gravity wave signatures (since many profiles have been averaged) such that any significant non-zero $E_P$-values must be artifacts due to shortcomings of the algorithm. The result of this exercise is shown in the middle panels of Figure 14. Quite obviously this analysis yields regions of very large apparent $E_P$-values in regions of the TTIL. Compared to the panels in the upper row of the figure, it is also clear that these artifacts actually dominate the $E_P$-values in the TTIL-region. In addition, we note that additional artifacts are observed at larger altitudes and also in other latitude and

altitude regions. However, for these, their absolute values are significantly less than in the data sets in the upper row such that the contribution of these artifacts to the overall $E_P$-values is not significant. This is also clearly seen in the lowermost panels of Figure 14 which show the difference of the full $E_P$-distribution (in the top row) and the contributions from the monthly mean zonal mean profiles (in the middle). In these 'corrected' $E_P$-distributions, the maximum values in the tropical TIL-region have basically disappeared whereas there is hardly any change visible at other altitude and latitude regions. Coming back to the right

panel of Figure 10 we hence conclude that the relatively large differences seen below 25km do not reflect real differences in terms of gravity wave activity in RO data and model data. Rather the differences are caused by differences in the representation of the TTIL and the difficulty to properly derive $E_P$-values in its environment from vertical profiles alone.





## 6 Summary and Conclusions

In this manuscript we compared operational METOP GPS-RO temperatures and derived gravity wave potential energy densities with corresponding ECMWF operational analysis and ERA Interim reanalysis data sets. This was done to answer two questions, namely whether the sampling and data quality of the operational RO data set is sufficient to properly characterize the global

gravity wave activity (measured in terms of $E_P$) on a monthly basis. Furthermore, we investigated whether the METOP-observations are consistent with the ECMWF model-fields such that the latter can be used for the interpretation of observational results.

For this purpose, we analyzed a total of 30 month of RO data for the period from July 2014 to December 2016. We calculated monthly mean temperatures and $E_P$-values on a grid of 5° in latitude, 10° in longitude, and at a vertical resolution of 1 km for

altitudes between 20 and 40 km. This was done for two RO data sets, namely for so-called 'dry' and 'wet' data both provided by EUMETSAT's ROM SAF. Dry temperatures are directly derived from refractivity profiles which in turn are estimated from bending angle observations with the GPS RO-technique. In contrast, wet temperatures are the result of a one dimensional variational retrieval that uses additional a priori information on atmospheric humidity and temperature from ECMWF model fields. Subsequently both temperatures and $E_P$-values from RO-observations and from ECMWF analysis and reanalysis model

fields were compared rigorously.

The comparison of temperatures showed very low systematic differences between RO dry temperatures and ECMWF model fields between 20 and 30 km (i.e., median temperature differences between -0,2 and +0,3 K) which then increased with height to yield median differences of +1,0 K at 34 km and +2,2 K at the maximum considered altitude of 40 km. Compared to this, median differences between RO-wet temperatures and ECMWF-model data were below 0,16 K for all considered altitudes,

which is as expected since ECMWF model data were used to constrain the RO data retrieval.

We then introduced a method to derive $E_P$ from temperature profiles by applying a fifth order Butterworth filter with cutoff wavelength of 15 km to both RO and model data. An initial comparison of $E_P$-time series in selected altitude ranges and at three selected locations in Sodankylä, Northern Scandinavia, in the German Bavarian Forest, and at Lauder, New Zealand, yielded overall very good agreement: below 35 km, this agreement was both very good in terms of seasonal variation and in terms

of absolute $E_P$-values. A striking result, however, was that for Northern Scandinavia - which is known as a region of strong orographic wave activity - the horizontally coarser resolved ERA-Interim data underestimated a large winter peak of $E_P$ that was present in both the RO data as well as in the higher resolution IFS-data. At altitudes above 35 km, however, both models did follow the observed seasonal variation of $E_P$ qualitatively but underestimated the observed values by about a factor of two. This is likely caused by the damping of small scale model structures by the model's sponge layer. The same $E_P$ time series

from RO-observations were then also compared to local Rayleigh lidar observations. This comparison showed a qualitatively similar seasonal $E_P$ variation with both experimental techniques but it also revealed that the RO-technique underestimates the locally observed values by about a factor of two. This low bias is likely caused by the very different observational filter of RO- and lidar observations where in particular the long line of sight of RO-observations that are carried out in limb geometry





severely hampers the detection of waves with horizontal wavelengths smaller than 190 - 270 km while the lidar observations are also sensitive to much smaller horizontal wavelengths.

Finally we compared the full 30 month data set of RO and model $E_P$-fields. The corresponding statistical analysis shows large correlation coefficients (0,4 - 1,0) between all considered data sets (RO dry, RO wet, ERA Interim, and IFS) for all altitudes between 20 and 40 km. A minimum correlation (of still 0,4) was found at altitudes around 28 km, where the ECMWF-anaylsis and reanalysis fields do not seem to capture the GW-activity that is observed in the RO data. The reason for this discrepancy could not be identified and should be investigated in a future study. Concerning absolute differences between observed and modelled $E_P$-values, the median difference was relatively small at all altitudes with an exceptional feature between 20 and 25 km where both the median difference between RO- and model-data increased and where also the corresponding variability was found to be very large. The reason for this was identified as an artifact in the $E_P$-algorithm: this erroneously interprets the pronounced climatological feature of the TTIL at latitudes between $\pm20°$ and altitudes between 20 and 25 km as gravity wave activity hence yielding a) very large $E_P$-values in this area and b) also large differences between model and observations because the RO data show a much more pronounced TTIL than IFS and ERA-Interim. Based on that finding we also suggested a correction for this effect based on an estimate of this 'artificial' $E_P$ using monthly mean zonal mean temperature profiles which do reveal a very pronounced TTIL but which should not contain any remaining GW-signatures due to strong averaging. We suggest to use this correction in the future to study, for example, the annual cycle of zonal mean GW activity using the here considered data.

In summary, our analysis shows very good quantitative agreement between monthly mean RO-dry and ERA-Interim and IFS-data in the altitude range between 20 - 40 km altitude. Hence, both research questions posed at the beginning of this study can be answered positively: for one, this very good agreement shows that METOP RO-dry data are a suitable data base to study monthly mean global gravity wave activity in the altitude range between 20 and 40 km (with the caveat that the tropical TIL-region needs to be considered with particular care). In addition, the good agrement between RO-dry and ECMWF data also implies that the combination of both appears to be a versatile combined data set for the study of processes determining the GW climatology. Future questions to be considered are, for example, in how far the strong stratospheric jet streams influence the observed GW morphology in the stratosphere. While model results of Dunkerton (1984) and more recently also Sato et al. (2009) and Sato et al. (2012) have long suggested that the waves should be refracted into the jet streams, observational evidence for this process based on global data is still scarce. This and other research questions will be investigated in future studies.

## 7   Data availability

RO data are availabe from www.romsaf.org. Details regarding access to ECMWF-data can be found under www.ecmwf.int.

*Author contributions.*  MR devised the study, wrote the data analysis code, performed the analysis, and drafted the 1st manuscript version. AD provided the ECMWF data and helped with its handling and analysis. BK provided the lidar data and contributed to its analysis. All authors contributed to writing the text.



*Competing interests.* The authors declare that they have no competing interests.

*Acknowledgements.* GPS RO data provision by EUMETSAT's ROM SAF is greatly appreciated. Access to the ECMWF data was possible through the special project "HALO Mission Support System". Part of this research was conducted within the scope of the German research initiative 'Role of the Middle Atmosphere in Climate (ROMIC)' under grant 01LG1206A provided by the German Ministry for Education

5    and Research. Partial funding was also provided by the German Science Foundation (DFG) via the research unit MS-GWaves (GW-TP/DO 1020/9-1, PACOG/RA 1400/6-1). MR would also like to thank J. Wickert for pointing his attention to METOP RO data and W. Beer for his efficient handling of file downloads from the ROM SAF. MR also thanks P. Preusse and M. Ern for valuable comments on early results achieved within this study and Natalie Kaifler for critically reading and commenting on the text.



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





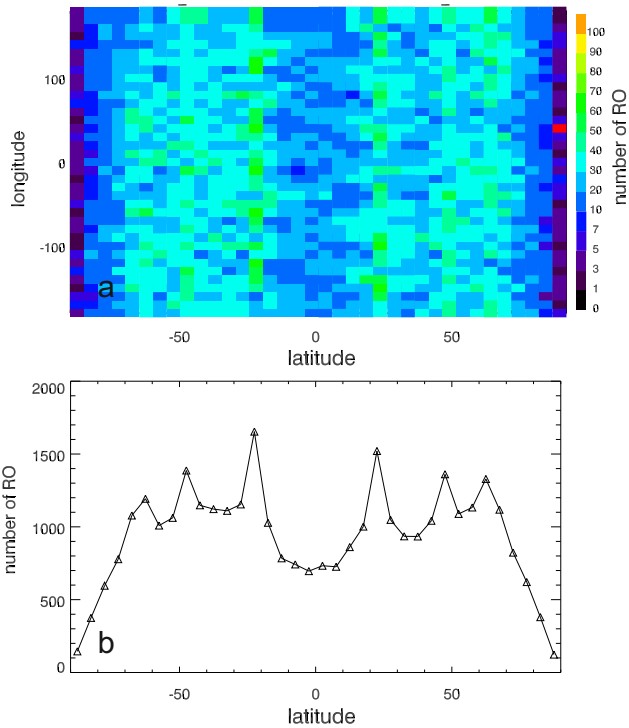

**Figure 1.** Panel a: Number of Metop A and B radio occultations per 5 degree latitude and 10 degree longitude bin in June 2015. The total number of occultations in this month is about 35.000. Panel b: Number of occultations per 5 degree latitude bin intergrated over all longitudes.



**Figure 2.** Zonal mean temperatures as a function of latitude and altitude for the months March, June and December 2015 (top to bottom) from Metop A/B radio occultations (left) and from ERA Interim (right).



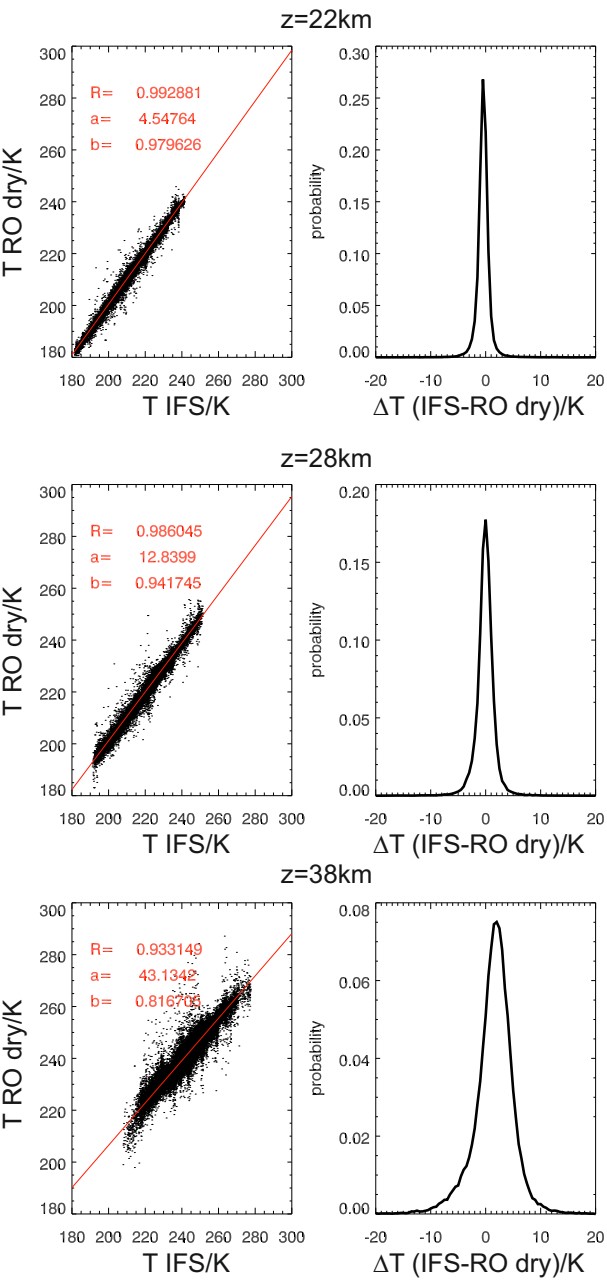

**Figure 3.** Scatter plots (left) between RO dry temperatures and corresponding IFS-data for 30 months of data between July 2014 and December 2016 for three selected altitudes. The red line shows a linear fit to the data with slope b, y-intercept a, and correlation coefficient R (see insert). Panels on the right show histograms of the corresponding temperature differences between IFS and RO dry data for the same selected altitudes.





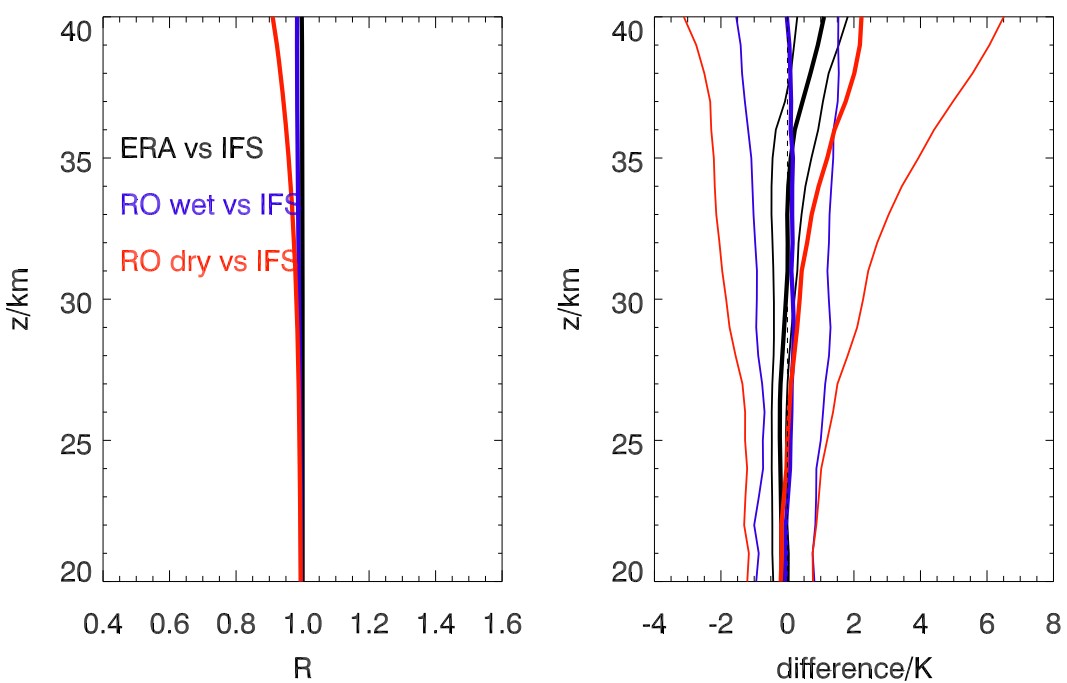

**Figure 4.** Left: Correlation coefficients as a function of altitude for the data the correlation between ERA Interim and IFS-data (black line), RO wet- and IFS-data (blue line), and RO-dry and IFS-data (red line). Right: Corresponding median temperature differences (thick lines) along with 10% and 90% percentiles (thin lines) as a function of altitude (same color code as in left panel).




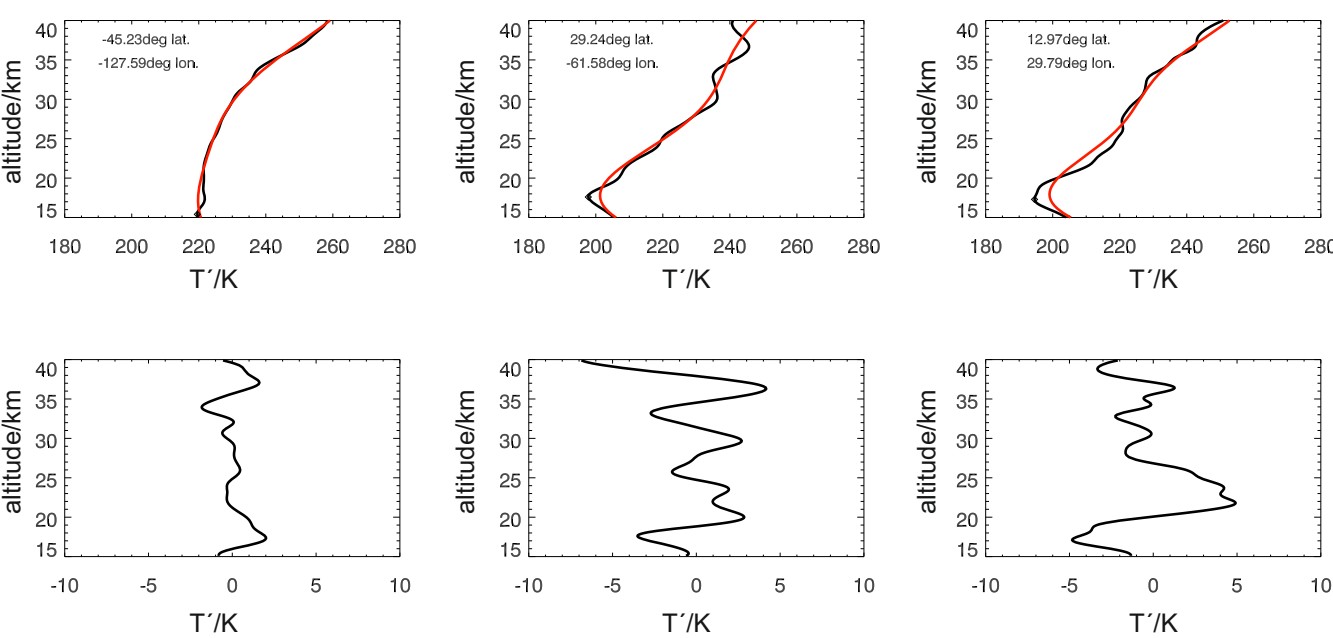

**Figure 5.** Upper panels: Sample radio occultation temperature profiles from December 2015 (black lines) with background profiles (red lines) as determined with a 5th order Butterworth filter with 15 km cuttoff wavelength following Ehard et al. (2015). Lower panels: corresponding temperature perturbation profiles (= radio occultation profile minus background profile).





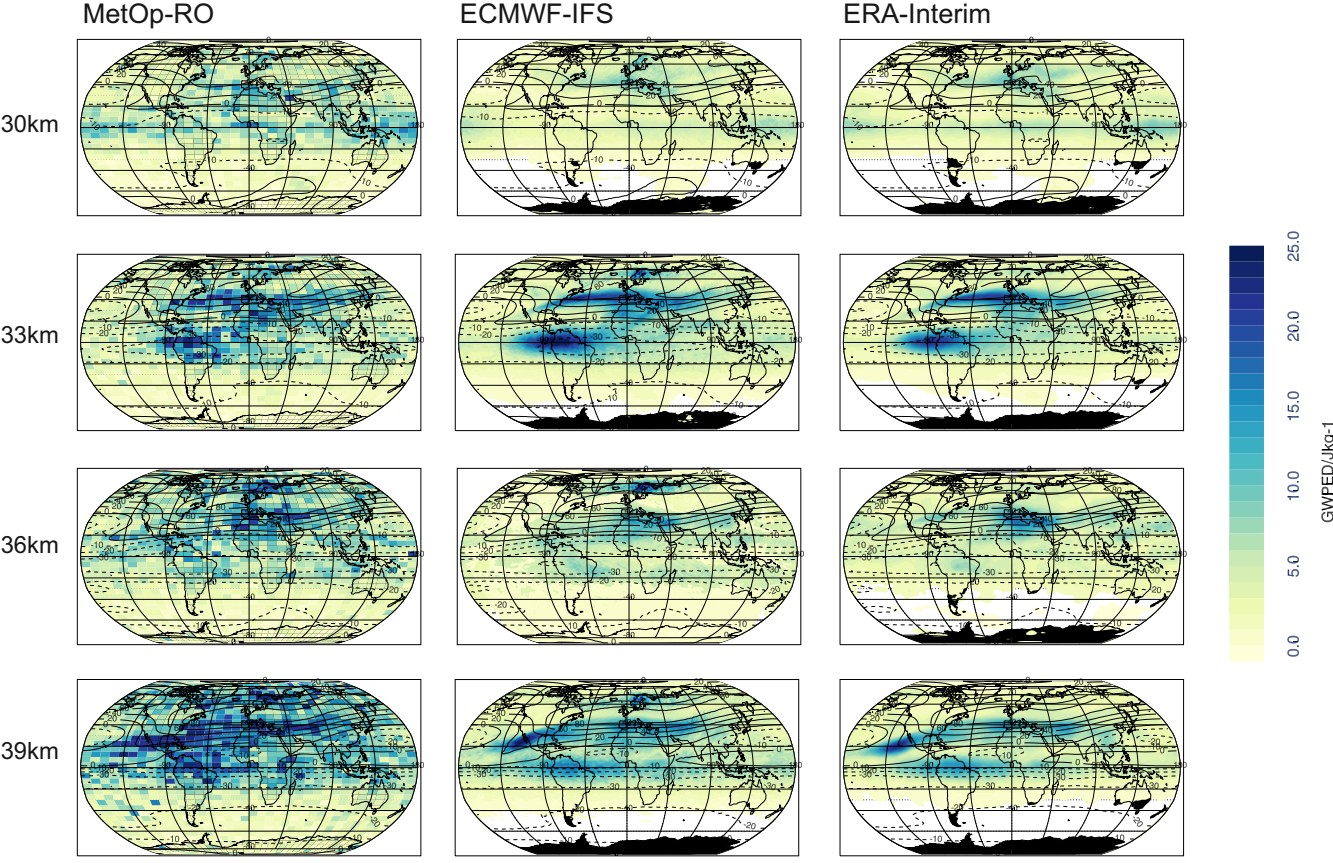

**Figure 6.** Monthly mean latitude-longitude cross sections of $E_p$ at selected altitudes of 30, 33, 36, and 39 km (from top to bottom) for December 2015. Left panels show METOP-RO dry-data, middle panels show IFS-data, and right panels show ERA-Interim data, respectively. In all panels, black contour lines show zonal wind values from ERA-Interim.





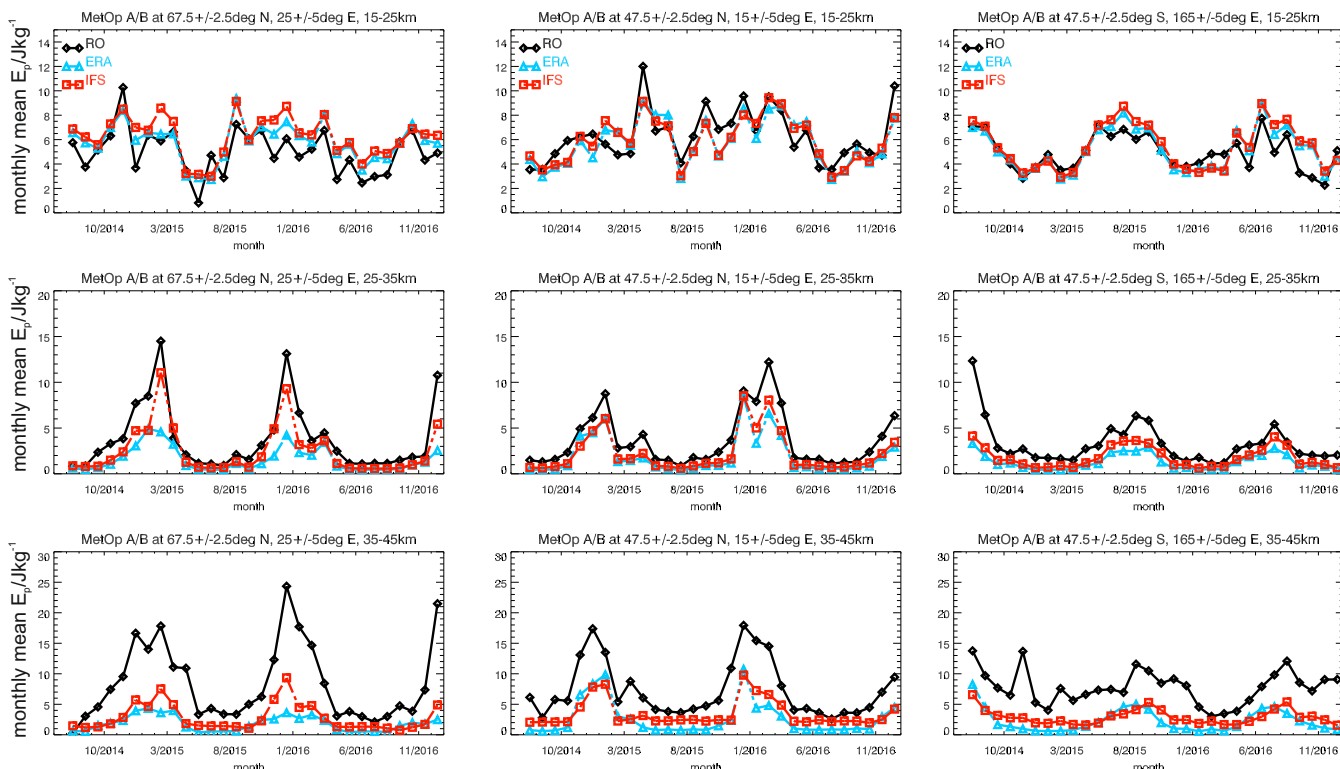

**Figure 7.** Comparison of time series of monthly mean $E_P$ from RO and model data (ERA-Interim and IFS) for three different locations: Sodankylä (left), Bavarian Forest (middle), and Lauder (right). The color code is explained in the insert. Panels shown in top, middle, and bottom row are for altitude ranges of 15-25 km, 25-35 km, and 35-45 km, respectively.



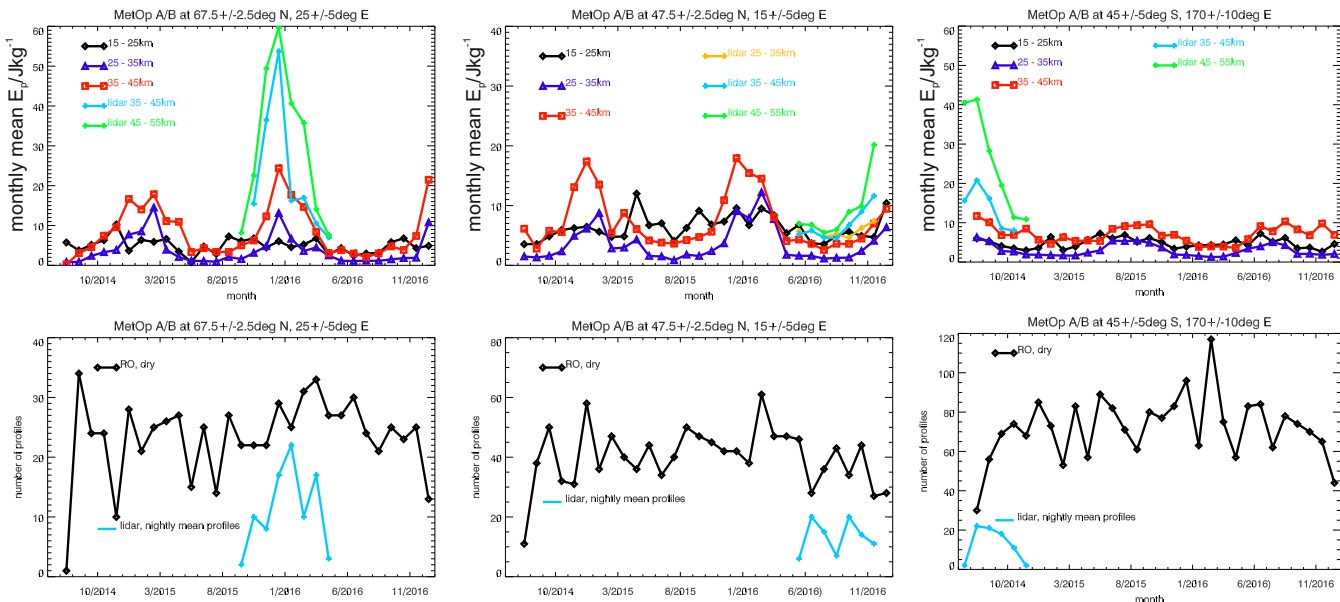

**Figure 8.** Upper panels: Comparison of time series of monthly mean $E_P$ from METOP-RO data for different altitude ranges (black, blue, and red curves; see insert for color code) with local Rayleigh lidar measurements of $E_P$ for the stations of Sodankylä (left), the Bavarian Forest (middle), and Lauder (right). Lidar $E_P$ are shown as yellow (25-35 km), light blue (35-45 km), or green (45-55 km) lines. Lower panels: Number of RO-profiles (black lines) and nightly mean lidar profiles (light blue lines) entering the monthly mean shown in the panels above.



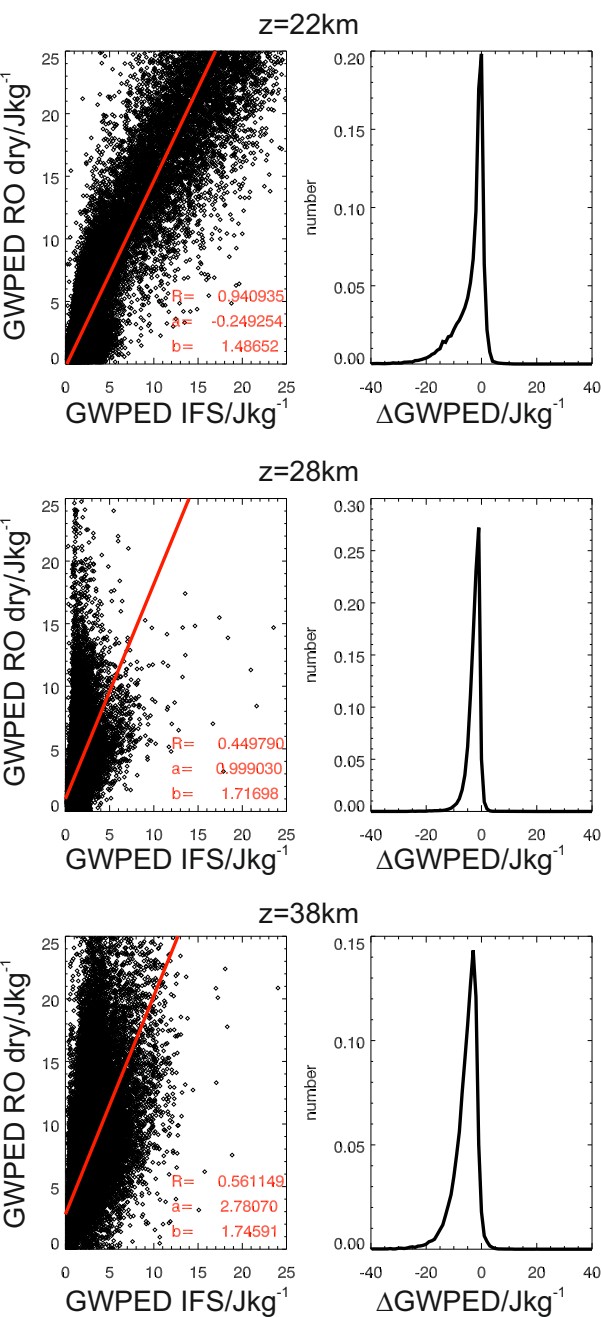

**Figure 9.** Left: Scatter plots between $E_P$-values derived from the IFS and RO-dry-data for three different altitudes, i.e., 22 km (top), 28 km (middle), and 38 km (bottom).The red line shows a linear fit to the data with slope b, y-intercept a, and correlation coefficient R (see insert). The panels on the right show corresponding histograms of the absolute difference between the two data sets.





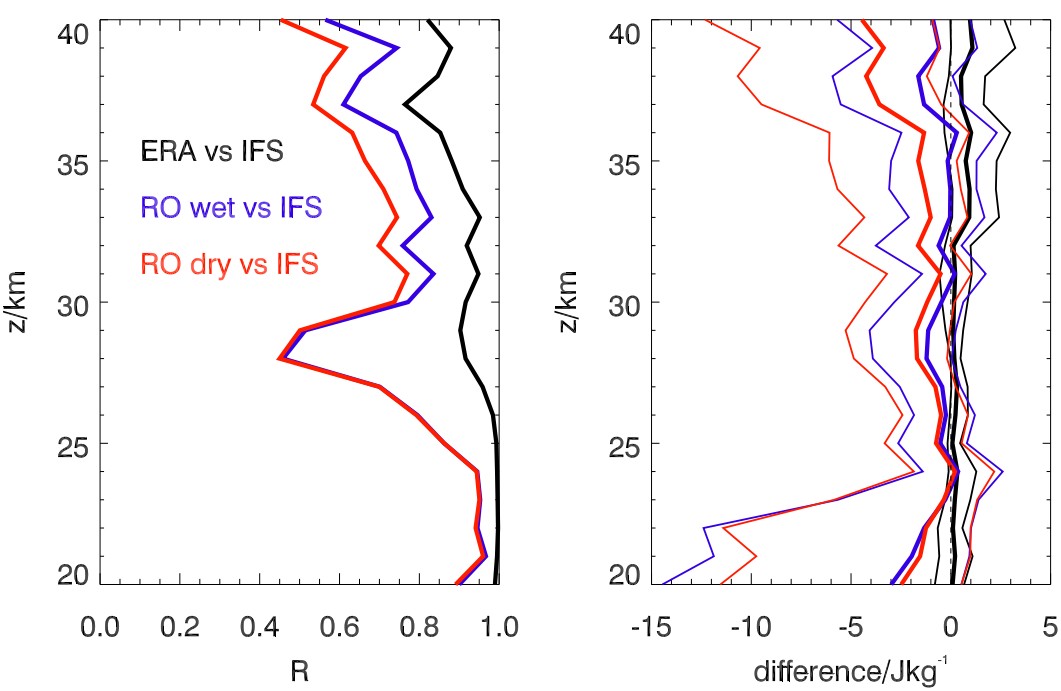

**Figure 10.** Left: Correlation coefficients as a function of altitude for the correlation between ERA Interim and IFS-data (black line), RO wet- and IFS-data (blue line), and RO-dry and IFS-data (red line). Right: Corresponding median temperature differences (thick lines) along with 10% and 90% percentiles (thin lines) as a function of altitude (same color code as in left panel).





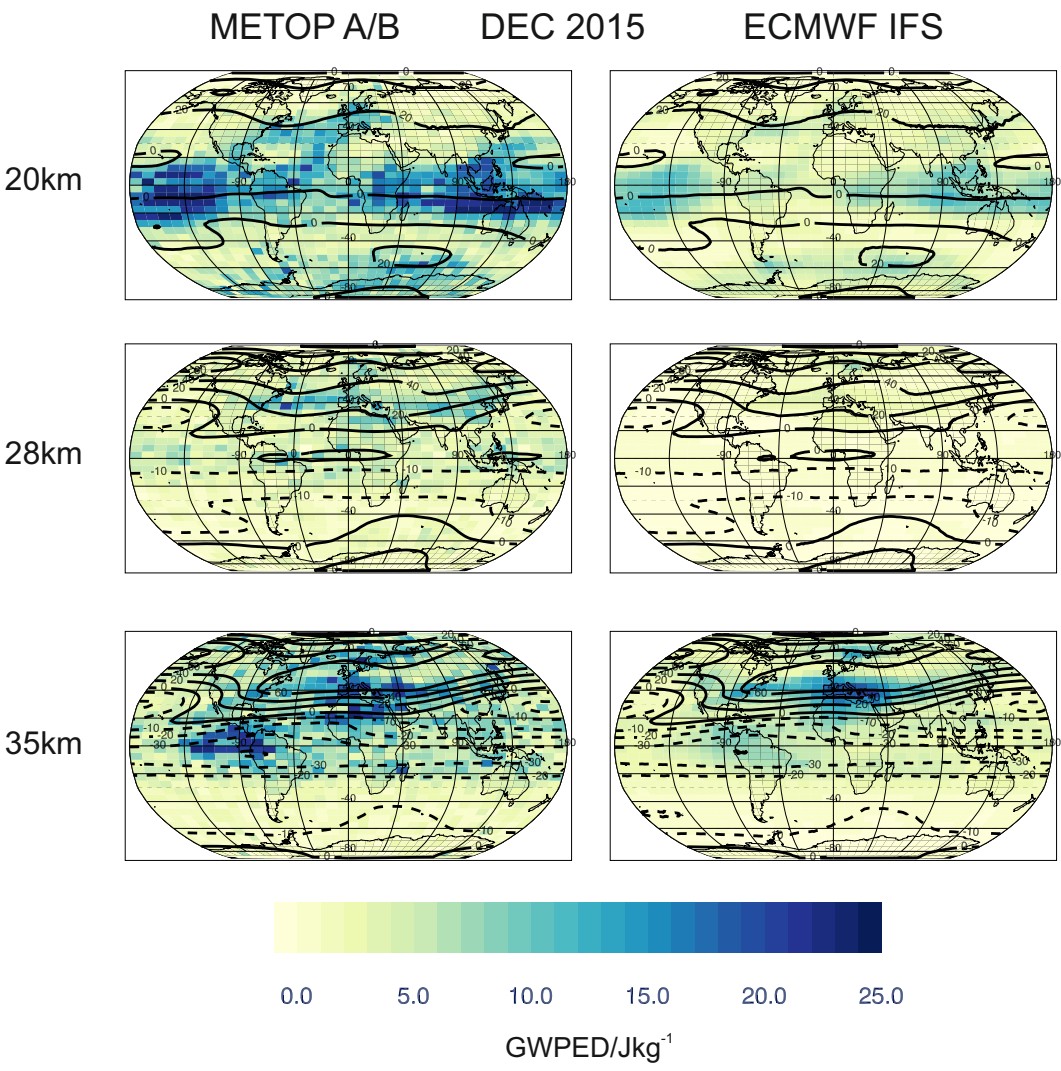

**Figure 11.** Left: Latitude-longitude distributions of $E_P$ based on GPS-RO-dry-data for December 2015 and altitudes of 20, 28, and 38 km (top to bottom). Right: Same as left panels but based on IFS data. In all panels black contours show zonal wind values from ERA-Interim.



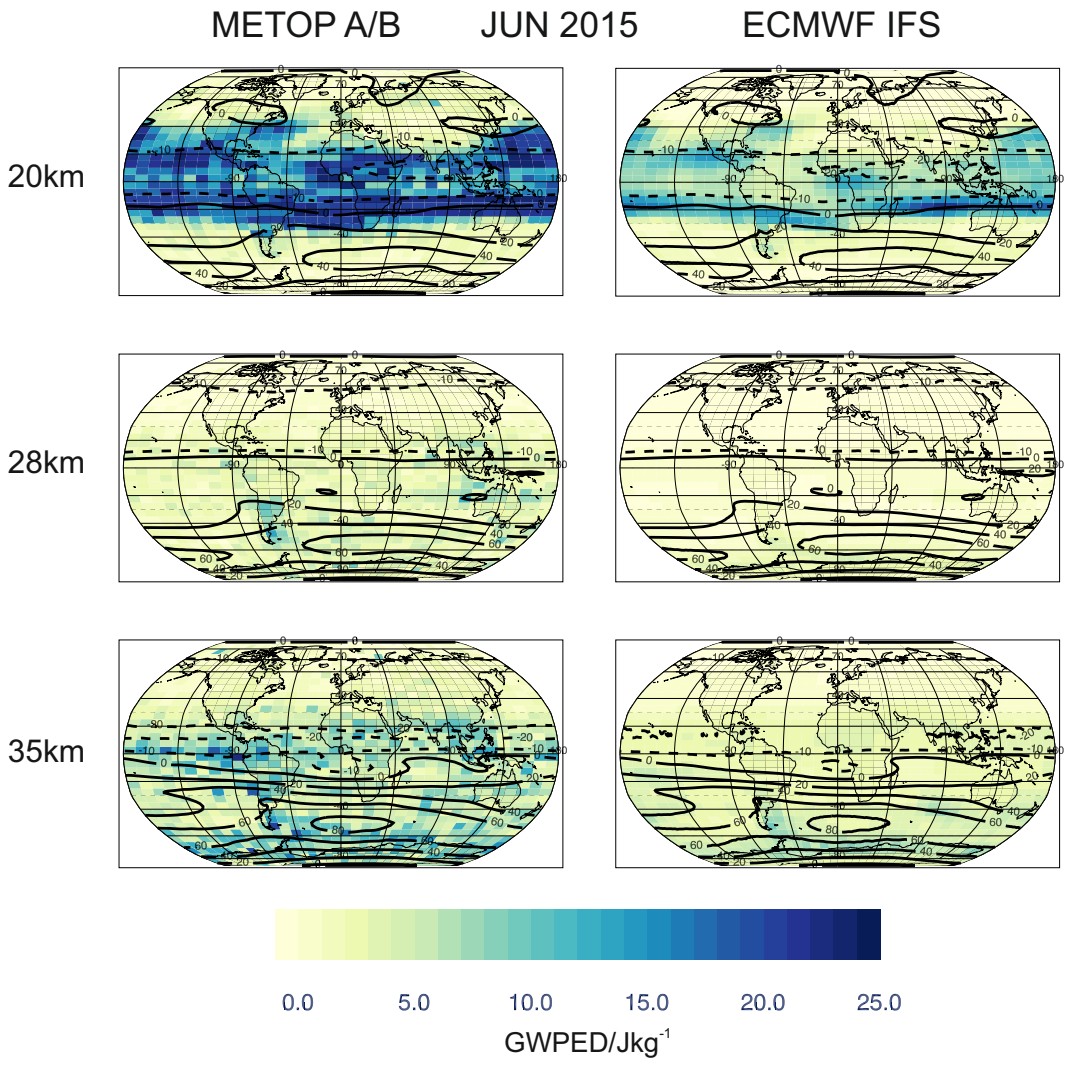

**Figure 12.** Same as Figure 11 but for June 2015.





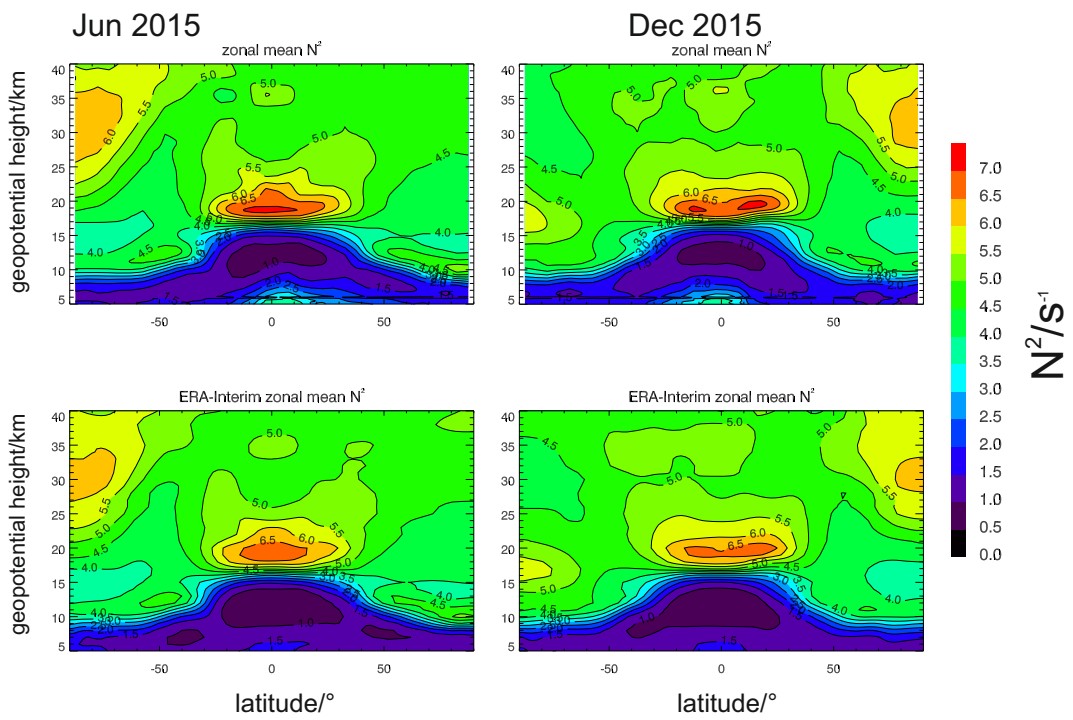

**Figure 13.** Zonal mean distribution of $N^2$ as a function of latitude and altitude for the months June 2015 (left) and December 2015 (right) based on GPS-RO-dry-data (upper panels) and ERA-Interim data (lower panels), respectively.





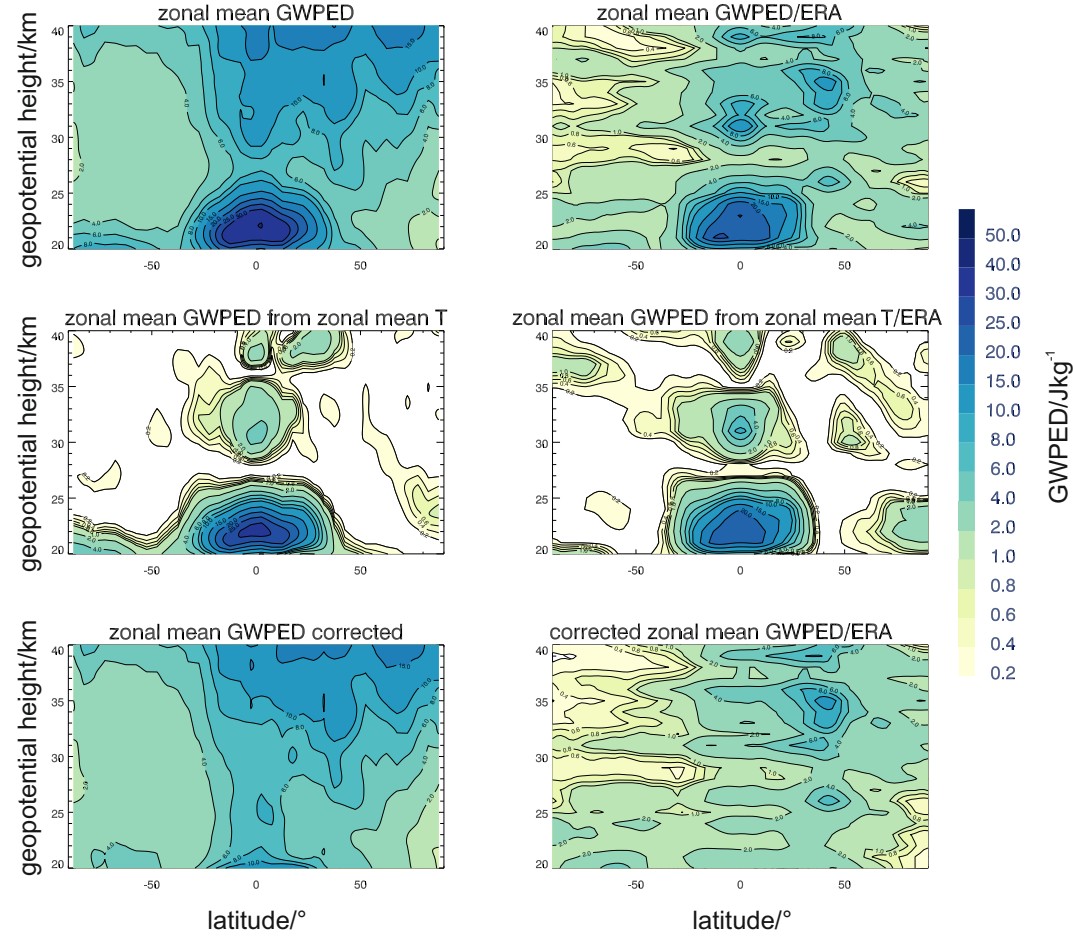

**Figure 14.** Upper panels: Monthly mean zonal mean distributions of $E_P$ as a function of latitude and altitude for December 2015 based on RO-dry-data (left) and ERA-Interim-data (right). Middle panels: Zonal mean apparent $E_P$-values derived from applying the $E_p$-algorithm to monthly mean zonal mean temperature profiles. Bottom: Difference between upper panels and middle panels.