# Peer review of "An intercomparison of stratospheric gravity wave potential energy densities from METOP GPS-radio occultation measurements and ECMWF model data"

_Atmospheric Measurement Techniques, 2017_

## Referee Comment (RC1) · Anonymous Referee #1 · 7 Nov 2017

This paper investigates whether the data set of METOP GPS radio occultations can be used for the analysis of gravity waves. Gravity wave potential energy derived from the radio occultations is compared to ECMWF IFS and ERA-Interim data. Qualitatively, good agreement is found for global distributions. Some shortcomings are discussed. For example, enhancements in the lower stratosphere in the tropics are attributed to effects of the tropopause, and a correction method for monthly averaged data is proposed. Comparison to gravity wave observations by lidar shows good correspondence in temporal variations. However, potential energies seen by lidar are higher by a factor

of two, which is explained by the better sensitivity of lidars for gravity waves of short horizontal wavelengths.

Overall, the paper shows that METOP GPS-radio occultations are a promising data set for gravity wave analysis. The paper is very well written, and publication in AMT is recommended after addressing my minor comments.

My main comment is that some more discussion is needed regarding the separation of temperature altitude profiles into background and temperature fluctuations due to gravity waves. The selected method is a vertical filter with cutoff at 15km vertical wavelength. The main idea is that all variations with wavelengths shorter than 15km are assumed to be gravity waves. This, however, is not stated clearly enough, and the pros and cons of this approach should be briefly discussed as detailed in my specific comments.

For specific and technical comments see below.

SPECIFIC COMMENTS:

(1) about Sect. 2.2: Please clarify that the spatial resolution mentioned in this section corresponds to the horizontal grid spacing. Atmospheric waves that are resolved by these data sets have scales that are considerably longer. According to Skamarock (2004) only scales exceeding the grid spacing by several times are resolved.

Skamarock, W. C.: Evaluating Mesoscale NWP Models Using Kinetic Energy Spectra, Mon. Wea. Rev., 132, 3019-3032, 2004.

(2) p.5, l.12, Sect. 2.3 Here, you call the reduced scatter of RO-wet temperatures a reduced "uncertainty range". Is this justified, or are RO-wet temperatures too smooth?

Above 30km RO-wet temperatures show a reduced scatter with respect to ECMWF IFS. However, at these altitudes the influence of a priori data should be increasing, and it is known that at high altitudes ECMWF is known to suffer from hyper-diffusion. Could it therefore happen that temperature fluctuations are suppressed in RO-wet temperatures because of an increasing influence of relatively smooth a priori data?

(3) p.5, l.28 The idea behind using a Butterworth filter should be stated more clearly, and shortcomings mentioned.

As far as I understand, variations in the vertical with scales longer than 15km are assumed to be the "background" (climatological structure plus planetary waves), while shorter scales are assumed to be fluctuations due to atmospheric gravity waves. This separation of scales will work well with a few exceptions. One exception is the tropopause region, which has been discussed in detail in the current paper. Another exception is the tropical stratosphere. While vertical wavelengths of planetary waves in the extratropics are quite long, this is different in the tropics where Kelvin waves usually have vertical wavelengths that are comparable to those of gravity waves. See also (5).

(4) p.5, l.28 Please mention that the use of the Butterworth filter in vertical direction has the advantage of being applicable in the same manner to all data sets considered, thus allowing a fair comparison.

(5) p.6 l.10-14 Epot close to the equator will be high-biased due to Kelvin waves

Kelvin waves in the tropics can have quite short vertical wavelengths, comparable to those of gravity waves. Kelvin waves can have considerable temperature variances of a few Kˆ2 on zonal average, and corresponding zonal average values of Epot could easily reach values of around 5 J/kg, which is non-negligible. This is particularly important because Fig.6 represents a case of tropical easterlies. Under these conditions the amplitudes of Kelvin waves will be amplified.

A climatology Kelvin waves in the tropical stratosphere is given, for example, in Ern et al. (2008)

Ern, M., Preusse, P., Krebsbach, M., Mlynczak, M. G., and Russell III, J. M.: Equatorial wave analysis from SABER and ECMWF temperatures, Atmos. Chem. Phys., 8, 845-869, 2008.

(6) p.9, about the Epot correction Do you think that monthly average temperatures are sufficient for deriving a correction, as proposed? Or may there be changes in the background on shorter time scales that would require averaging over shorter time intervals? Of course, this may be beyond the scope of the current paper, but should be carefully considered before making this correction operational.

see also p.11, l.16+17

(7) p.11, l.22 Please include the information that Kelvin waves could produce a high-bias of Epot in the tropics.

TECHNICAL COMMENTS:

p.2, l.7 GW are a major means to couple the -> GW are an important mechanism that couples the

p.2, l.30 please correct: ECMWF = "European Centre for Medium-Range Weather Forecasts"

Fig.1a Is the red dot at 90N an artifact, or is this a real accumulation of ROs?

p.3, l.18 The expression in parentheses is confusing; suggestion

a corresponding gridding (i.e., 36 × 36 5deg latitude × 10deg longitude bins) -> a corresponding gridding of 36 x 36 grid points (i.e., 5deg latitude × 10deg longitude bins)

p.4, l.8 Please check whether it is correct that T1279 corresponds to a horizontal resolution of 8km

To my knowledge, the ECMWF grid uses 2560 points at the equator, corresponding to 15km grid spacing. The numbers of T255/80km that given for ERA-Interim should however be correct.

p.4, l.12 from -> starting from

p.7, l.23 wavelengths if the phase fronts are perpendicular to the line of sight, -> wavelengths than is the case if the phase fronts are perpendicular to the line of sight,

p.8, l.33 cleat -> clear

p.13, l.33 stratopsphere -> stratosphere

p.13, l.34: Ern et al., 2016 -> Ern et al., 2017

p.14, l.12: LOVE -> Love ??

p.14, l.15 Hei, H., T., T. T., and Hirooka: -> Hei, H., Tsuda, T., and Hirooka, T.:

---

## Referee Comment (RC2) · Anonymous Referee #2 · 13 Nov 2017

The paper by Rapp et al. addresses stratospheric gravity wave (GW) activity derived from different data sets: GPS-RO temperature profiling by GRAS onboard Metop A/B, ECMWF operational analysis, ERA-Interim reanalysis as well as Rayleigh lidar measurements at specific locations. The central subject of this study is the intercomparison of GW potential energy densities derived from 'dry' and 'wet' GPS-RO retrievals, ECMWF IFS and ERA-Interim. Generally, a good agreement is found between GW potential energy distributions derived from these data sets, which is not totally surprising given that ECMWF model assimilates GPR-RO 'dry' measurements with a high level of

confidence, whereas GPS-RO 'wet' retrieval uses ECMWF data as a priori information. Nevertheless, some important discrepancies between RO- and model-based GW Ep fields are pointed out. The interpretation of the data sets is well set in the scientific background of previous publications and largely sound. A few exceptions are listed below. These points need to be corrected. The paper is generally well to read and recommended for publication in AMT after some revisions.

General remarks.

1) Choice of the useful vertical range for GPS-RO.

The choice of lower boundary of 20 km is conditioned by potential aliasing of tropopause layer's structure and variability as GW-induced perturbations. Although this excludes the entire extra-tropical lower stratosphere from the analysis, it allows to consider and compare the global distributions of GW Ep without heavy stipulations regarding the effects of tropical tropopause. My concern here mostly regards the upper boundary of 40 km. GPS-RO is a powerful tool for temperature profiling, which is why a good knowledge of actual capacities of this technique is crucial for atmospheric community. The majority of RO-based GW studies restrict the analysis to altitudes below 35 km because the uncertainty and the noise become too large at high altitudes. While this is confirmed by the results of this paper, I would love to see among the conclusions some more definitive statements regarding the usefulness of z>35 km RO data for GW retrieval.

2) Choice of Ep derivation method (Sect. 3).

As mentioned in the paper, there is a number of techniques for isolating the GW-induced perturbations in temperature profiles from the background atmospheric state. Here the authors opted to use a vertical detrending method based on a Butterworth filter. The advantage of the vertical detrending is that it can be applied to both local and global observations, enabling direct RO-lidar comparison. At that, I wonder how different would the results be for RO and ECMWF (particularly the TTIL issue) had the

authors used a horizontal detrending method for GW Ep retrieval, which seems to better handle the lower stratosphere region (Schmidt et al., 2016; Khaykin et al., 2016). Indeed, when the authors subtract the zonal-mean Ep profiles (which is already some sort of horizontal detrending), the TTIL anomaly disappears. A recommendation to use this correction is put forth in the conclusions but I wonder, wouldn't it be better just to use one of the horizontal detrending methods instead? I believe the authors should better justify the choice of Ep derivation method in consideration of its shortcomings before recommending it for future use.

Specific remarks.

P4, L17-19. Although the correct references are in place, the fact that the compared data sets are not independent requires some more specific information on the assimilation of RO data in ECMWF IFS and ERA-Interim.

P6, L20-22. It is indeed surprising that ERA doesn't see the Scandinavian GW hotspot. The validity of the proposed explanation (invoking coarser resolution of ERA) could be verified by examining global Ep distribution for June, when the strong Patagonian GW hot spot is well pronounced.

P.7, L17-31. What is missing in this discussion is the mention of the large difference in vertical resolution of GPS-RO and lidar at high altitudes.

P7, L33-34. The sentence could be reformulated to make it clearer that it is the derived Ep values that are low-biased and not the actual temperature data.

P8, L27. The relatively large bias is rather seem below 23 km.

P.9,L.10-11. What is somewhat controversial here is that the higher N2 values within the TTIL derived from RO should lead to lower Ep, whereas the results suggest the opposite. It is understandable that RO should better resolve the fluctuations, but invoking the differences in N2 field in this context should be done more carefully.

Technical remarks.

P.5,L9. Remove "km" after the right parenthesis

P.8, L.31. "At these altitudes" => at the level of lowest correlation?

P.9, L24. "larger altitudes" => higher altitudes?

Throughout Sect. 6, comma is erroneously used as a decimal separator instead of a point.

Fig. 3. The data are missing in the left-hand panels.

Fig. 3 and 9. The X-axis of right-hand panels could be reduced to enhance the readability of the histograms.

Fig. 4 left panel. The X-axis scale could be reduced to say 0.8 – 1.1

Fig. 5 upper panels. X-axis caption should be T and not T'

Fig. 6. The black-shaded land areas whenever Ep values are too low are somewhat confusing.
* * *

---

## Author Comment (AC1) · 22 Dec 2017

**Final and cumulative response to the reviewers' comments on our manuscript**

"An intercomparison of stratospheric gravity wave potential energy densities from METOP GPS-radio occultation measurements and ECMWF model data"

by M. Rapp, A. Dörnbrack, and B. Kaifler

submitted for publication in Atmospheric Measurement Techniques (amt-2017-346)

Weßling, 22.12.2017

We greatly appreciate the reviewers' positive assessment of our work as well as their constructive comments. For the revised version, all comments have been addressed. In the following we respond to all comments point by point. For clarity, we first repeat the comments of the referees (in black), we then respond to these (in blue), and then indicate the changes made to the text where appropriate (in red).

**Reviewer 1**

[..] Overall, the paper shows that METOP GPS-radio occultations are a promising dataset for gravity wave analysis. The paper is very well written, and publication in AMT is recommended after addressing my minor comments.

We are grateful to the reviewer for this encouraging judgment.

My main comment is that some more discussion is needed regarding the separation of temperature altitude profiles into background and temperature fluctuations due to gravity waves. The selected method is a vertical filter with cutoff at 15 km vertical wavelength. The main idea is that all variations with wavelengths shorter than 15 km are assumed to be gravity waves. This, however, is not stated clearly enough, and the pros and cons of this approach should be briefly discussed as detailed in my specific comments.

This is an excellent point raised by both reviewers. As a consequence, we have revised the text in several places as indicated below in response to the more specific points. In addition, mainly in response to reviewer 2, we have added an additional figure (Figure 15) in the revised manuscript in which we compare latitude-altitude distributions of monthly mean zonal mean fields of $E_P$ derived with the standard method applied in this study (i.e., using a fifth order Butterworth filter with a cutoff at 15 km wavelength in the vertical) as well as derived using a horizontal background determination method. Similarities and differences from both analysis techniques are discussed and a recommendation for future work is formulated (see also our detailed answer to comment 6 below).

Specific comments

(1) about Sect. 2.2: Please clarify that the spatial resolution mentioned in this section corresponds to the horizontal grid spacing. Atmospheric waves that are resolved by these data sets have scales that are considerably longer. According to Skamarock (2004) only scales exceeding the grid spacing by several times are resolved. Skamarock, W. C.: Evaluating Mesoscale NWP Models Using Kinetic Energy Spectra, Mon. Wea. Rev., 132, 3019-3032, 2004.

Thanks for pointing this out. We have corrected the text accordingly.

[..] These have a horizontal grid spacing of 16 km ($T_L1279$) and were evaluated on 25 pressure levels between 1000 and 1 hPa which we converted to geopotential heights and interpolated them on a regular vertical grid with 1 km spacing. We note that according to Skamarock (2004) only scales exceeding the grid spacing by several times are resolved.

The horizontal grid spacing of the data set is approximately 80 km (T255).

(2) p.5, l.12, Sect. 2.3 Here, you call the reduced scatter of RO-wet temperatures a reduced "uncertainty range". Is this justified, or are RO-wet temperatures too smooth? Above 30km RO-wet temperatures show a reduced scatter with respect to ECMWF IFS. However, at these altitudes the influence of a priori data should be increasing, and it is known that at high altitudes ECMWF is known to suffer from hyper-diffusion. Could it therefore happen that temperature fluctuations are suppressed in RO-wet temperatures because of an increasing influence of relatively smooth a priori data?

The reviewer again has a good point here. In order to give a more neutral description of Figure 4 we have changed the wording "uncertainty range" to "variability range".

(3) p.5, l.28 The idea behind using a Butterworth filter should be stated more clearly, and shortcomings mentioned. As far as I understand, variations in the vertical with scales longer than 15km are assumed to be the "background" (climatological structure plus planetary waves), while shorter scales are assumed to be fluctuations due to atmospheric gravity waves. This separation of scales will work well with a few exceptions. One exception is the tropopause region, which has been discussed in detail in the current paper. Another exception is the tropical stratosphere. While vertical wavelengths of planetary waves in the extratropics are quite long, this is different in the tropics where Kelvin waves usually have vertical wavelengths that are comparable to those of gravity waves. See also (5).

(4) p.5, l.28 Please mention that the use of the Butterworth filter in vertical direction has the advantage of being applicable in the same manner to all data sets considered, thus allowing a fair comparison.

(5) p.6 l.10-14 Epot close to the equator will be high-biased due to Kelvin waves Kelvin waves in the tropics can have quite short vertical wavelengths, comparable to those of gravity waves. Kelvin waves can have considerable temperature variances of a few K^2 on zonal average, and corresponding zonal average values of Epot could easily reach values of around 5 J/kg, which is non-negligible. This is particularly important because Fig.6 represents a case of tropical easterlies. Under these conditions the amplitudes of Kelvin waves will be amplified. A climatology Kelvin waves in the tropical stratosphere is given, for example, in Ern et al. (2008) Ern, M., Preusse, P., Krebsbach, M., Mlynczak, M. G., and Russell III, J. M.: Equatorial wave analysis from SABER and ECMWF temperatures, Atmos. Chem. Phys., 8, 845- 869, 2008.

Reply to 3, 4, and 5: We agree that it is critical to better explain the idea behind using a Butterworth filter. Also the limitations of this approach in the tropics must be more clearly mentioned and it needs to be pointed out more directly that the most important advantage of this approach is that all four data sets (including the ground based lidar data) can be analyzed with identical analysis routines. The text has been changed as follows:

For this study, we follow the approach of Ehard et al. (2015), i.e., we apply a fifth-order Butterworth filter with a cutoff wavelength of 15 km to vertical temperature profiles from the RO-measurements, from ERA Interim, from the IFS, as well as from ground based lidar -measurements. Applying this filter to altitude profiles implies that scales longer than 15 km are assumed to be the "background" (climatological structure plus planetary waves), denoted $T_0(z)$, while shorter scales are assumed to be fluctuations due to atmospheric gravity waves. This separation is expected to work well except for the tropical stratosphere where Kelvin waves are known to occur with vertical wavelengths well below 15 km (e.g., Ern et al., 2008; Randell and Wu, 2005). Hence, $E_P$ must be expected to be biased high in the tropics.
Nevertheless we stick to this approach since it has the advantage that all data sets analyzed in this study can be treated with identical analysis routines thus allowing us to directly and quantitatively compare $E_P$-values from four independent data sets.

(6) p.9, about the Epot correction: Do you think that monthly average temperatures are sufficient for deriving a correction, as proposed? Or may there be changes in the background on shorter time scales that would require averaging over shorter time intervals? Of course, this may be beyond the scope of the current paper, but should be carefully considered before making this correction operational. See also p.11, l.16+17

We agree with both reviewers that this correction indeed warrants a more in-depth discussion. In the new Figure 15 we show a monthly mean zonal mean distribution of $E_P$ that was derived using a horizontal background determination method. In the same figure we further show monthly mean zonal mean fields of vertical kinetic energies, i.e., $VE=0.5*w^2$, due to gravity waves (see e.g., Geller and

Gong, 2010). Note that the mean vertical velocity is zero so that non-zero VE-structures are mainly due to gravity waves. This comparison clearly shows that $E_P$-values derived with the horizontal background derivation agree much better in terms of their morphology to vertical kinetic energy values than $E_P$-values determined from vertical profiles (compare Figures 14 and 15). Figure 15 further shows that the correction method improves the qualitative comparison between $E_P$ and VE but that it cannot eliminate all features that are apparently not due to gravity waves. Closer inspection of the data sets reveals that this is partly because some of the non-gravity wave structures (mainly the TTIL) are not zonally homogeneous such that correcting for them using zonal mean fields cannot eliminate the corresponding signatures completely.

In the context of Figure 15, the following text has been added:

[revised manuscript text omitted]

(7) p.11, l.22 Please include the information that Kelvin waves could produce a high bias of Epot in the tropics.

Done.

TECHNICAL COMMENTS:
p.2, l.7 GW are a major means to couple the -> GW are an important mechanism that couples the

Changed as suggested.

p.2, l.30 please correct: ECMWF = "European Centre for Medium-Range Weather Forecasts"

Thanks. Changed as suggested.

Fig.1a Is the red dot at 90N an artifact, or is this a real accumulation of ROs?

It is indeed an artifact and has been removed.

p.3, l.18 The expression in parentheses is confusing; suggestion a corresponding gridding (i.e., 36 x 36 5deg latitude x 10deg longitude bins) -> a corresponding gridding of 36 x 36 grid points (i.e., 5deg latitude x 10deg longitude bins)

Thanks. Changed as suggested.

p.4, l.8 Please check whether it is correct that T1279 corresponds to a horizontal resolution of 8km.To my knowledge, the ECMWF grid uses 2560 points at the equator, corresponding to 15km grid spacing. The numbers of T255/80km that given for ERA-Interim should however be correct.

The reviewer is correct. We have corrected the grid spacing to 16 km.

p.4, l.12 from -> starting from

We corrected the typo.

p.7, l.23 wavelengths if the phase fronts are perpendicular to the line of sight, -> wavelengths than is the case if the phase fronts are perpendicular to the line of sight,

Thanks. Changed as suggested.

p.8, l.33 cleat -> clear

Corrected.

p.13, l.33 stratopsphere -> stratosphere

Corrected.

p.13, l.34: Ern et al., 2016 -> Ern et al., 2017

Corrected.

p.14, l.12: LOVE -> Love ??

Corrected.

p.14, l.15 Hei, H., T., T. T., and Hirooka: -> Hei, H., Tsuda, T., and Hirooka, T.:

Corrected. Thanks for the careful reading!

---

## Author Comment (AC2) · 22 Dec 2017

**Final and cumulative response to the reviewers' comments on our manuscript**

"An intercomparison of stratospheric gravity wave potential energy densities from METOP GPS-radio occultation measurements and ECMWF model data"

by M. Rapp, A. Dörnbrack, and B. Kaifler

submitted for publication in Atmospheric Measurement Techniques (amt-2017-346)

Weßling, 22.12.2017

We greatly appreciate the reviewers' positive assessment of our work as well as their constructive comments. For the revised version, all comments have been addressed. In the following we respond to all comments point by point. For clarity, we first repeat the comments of the referees (in black), we then respond to these (in blue), and then indicate the changes made to the text where appropriate (in red).

**Reviewer 2**

[..]The interpretation of the data sets is well set in the scientific background of previous publications and largely sound. A few exceptions are listed below. These points need to be corrected. The paper is generally well to read and recommended for publication in AMT after some revisions.

Thanks very much for the careful review and the encouraging judgement of our work.

General remarks.

1) Choice of the useful vertical range for GPS-RO.
The choice of lower boundary of 20 km is conditioned by potential aliasing of tropopause layer's structure and variability as GW-induced perturbations. Although this excludes the entire extra-tropical lower stratosphere from the analysis, it allows to consider and compare the global distributions of GW Ep without heavy stipulations regarding the effects of tropical tropopause. My concern here mostly regards the upper boundary of 40 km. GPS-RO is a powerful tool for temperature profiling, which is why a good knowledge of actual capacities of this technique is crucial for atmospheric community. The majority of RO-based GW studies restrict the analysis to altitudes below 35 km because the uncertainty and the noise become too large at high altitudes. While this is confirmed by the results of this paper, I would love to see among the conclusions some more definitive statements regarding the usefulness of z>35 km RO data for GW retrieval.

At the end of Section 3 we had already written: "For this reason, we will exclude altitudes below 20 km from our analysis and focus on the altitude range between 20 and 40 km only, knowing, of course, that the largest altitudes need to be treated with care since noise of RO-data is known to pick up significantly above 35 km altitude (Marquardt and Healy, 2005)."

We do agree with the reviewer that our analyses in Section 4 do confirm that the actual useful range of data is below 35 km. This has now also been explicitly pointed out in the conclusions.

Also, it is well known that noise in RO-data picks up substantially above 35 km such that several previous studies have recommended to restrict the useful range of RO-data for GW analysis to below 35 km (e.g., Schmidt et al., 2008). This previous recommendation is clearly supported by our analysis.

2) Choice of Ep derivation method (Sect. 3).
As mentioned in the paper, there is a number of techniques for isolating the GW-induced perturbations in temperature profiles from the background atmospheric state. Here the authors opted to use a vertical detrending method based on a Butterworth filter. The advantage of the vertical detrending is

that it can be applied to both local and global observations, enabling direct RO-lidar comparison. At that, I wonder how different would the results be for RO and ECMWF (particularly the TTIL issue) had the authors used a horizontal detrending method for GW Ep retrieval, which seems to better handle the lower stratosphere region (Schmidt et al., 2016; Khaykin et al., 2016). Indeed, when the authors subtract the zonal-mean Ep profiles (which is already some sort of horizontal detrending), the TTIL anomaly disappears. A recommendation to use this correction is put forth in the conclusions but I wonder, wouldn't it be better just to use one of the horizontal detrending methods instead? I believe the authors should better justify the choice of Ep derivation method in consideration of its shortcomings before recommending it for future use.

As already pointed out in our response to reviewer 1, this is an excellent and important comment. In order to address it, we have pointed out more clearly in the revised manuscript that we are using a vertical detrending method in order to treat all data sets with the same analysis procedure such that derived EP-values are directly comparable. The text added in the revised manuscript in response of this comment as well as to comment 5 of reviewer 1 is as follows:

For this study, we follow the approach of Ehard et al. (2015), i.e., we apply a fifth-order Butterworth filter with a cutoff wavelength of 15 km to vertical temperature profiles from the RO-measurements, from ERA Interim, from the IFS, as well as from ground based lidar measurements. [..] we stick to this approach since it has the advantage that all data sets analyzed in this study can be treated with identical analysis routines thus allowing us to directly and quantitatively compare $E_P$-values from four independent data sets.

In addition, we have added Figure 15 with its corresponding discussion (repeated below from our response to reviewer 1):

[revised manuscript text omitted]

Specific remarks.
P4, L17-19. Although the correct references are in place, the fact that the compared data sets are not independent requires some more specific information on the assimilation of RO data in ECMWF IFS and ERA-Interim.

We have now pointed out more clearly that the data sets are not completely independent.

Thus, ECMWF-model fields and METOP RO-data are obviously not completely independent.

P6, L20-22. It is indeed surprising that ERA doesn't see the Scandinavian GW hotspot. The validity of the proposed explanation (invoking coarser resolution of ERA) could be verified by examining global Ep distribution for June, when the strong Patagonian GW hot spot is well pronounced.

This is a very good suggestion. As suggested we have inspected the $E_P$-distributions for June 2015 in the Southern hemisphere based on both ERA and IFS-data. While IFS-data do show moderate EP-values over the Patagonian hot spot at 35 km altitude (see lower right panel in Figure 12), ERA-data don't as shown below.

[Figure]

This supports our hypothesis that the coarser resolution of ERA-data leads to the inability to reproduce localized mountain wave activity. A corresponding short statement has been added to the text.

Note that we have checked this interpretation by also comparing Ep -distributions over the well-known Patagonian GW-hot spot for June 2015. While METOP and IFS-data show clear signatures of strong GW activity in this region (see Figure 12), ERA-Interim again misses to reproduce this GW activity (not shown).

P.7, L17-31. What is missing in this discussion is the mention of the large difference in vertical resolution of GPS-RO and lidar at high altitudes. P7, L33-34. The sentence could be reformulated to make it clearer that it is the derived Ep values that are low-biased and not the actual temperature data.

The vertical resolution of the two techniques is actually not so much different: 900m for the lidar and ~1.5 km for the RO-data (Kaifler et al., 2005; Kursinski et al., 1997). This information has been added to the text. Also the sentence has been reformulated to make clear that Ep is low-biased and not T.

P8, L27. The relatively large bias is rather seen below 23 km.

This is correct and has been changed in the text.

P.9,L.10-11. What is somewhat controversial here is that the higher N2 values within the TTIL derived from RO should lead to lower Ep, whereas the results suggest the opposite. It is understandable that RO should better resolve the fluctuations, but invoking the differences in N2 field in this context should be done more carefully.

Here, one may not confuse the $N^2$-values shown in Figure 13 and the $N^2$-values used in the computation of Ep. For Figure 13, the $N^2$-values were calculated from monthly mean zonal mean T-profiles thus containing information on the climatological temperature profile. For Ep, however, $N^2$-values are calculated from the filtered individual temperature profiles such that scales smaller than 15 km (such as the TTIL) are suppressed. Hence the $N^2$-values shown in Figure 13 are an illustration of the climatological small scale structure in the temperature profile but do not enter the Ep-calculation. This has now been clarified in the text.

Note that the $N^2$-values in Figure 13 were computed from monthly mean zonal mean-temperatures. These must not be confused with the $N^2$-values used in our EP -calculation which is based on $T_0$-profiles. Remember that $T_0$-profiles result from filtering individual temperature profiles with a 5th-order Butterworth-filter with cutoff wavelength at 15 km such that $T_0$-profiles only contain spatial scales larger than 15 km, and hence do not contain information on the TTIL.

Technical remarks.

5,L9. Remove "km" after the right parenthesis

Done.

P.8, L.31. "At these altitudes" => at the level of lowest correlation?

We changed the wording to "at the altitude levels of lowest correlation"

P.9, L24. "larger altitudes" => higher altitudes? Throughout Sect. 6, comma is erroneously used as a decimal separator instead of a point.

We changed the wording to "higher altitudes". Also, we consistently replaced the commas by points whenever used as a decimal separator throughout Section 6.

Fig. 3. The data are missing in the left-hand panels.

This must have been a problem of the reviewer's pdf-file or pdf viewer. The one we see on the AMT-website does show the data.

Fig. 3 and 9. The X-axis of right-hand panels could be reduced to enhance the readability of the histograms.

Done as suggested.

Fig. 4 left panel. The X-axis scale could be reduced to say 0.8 – 1.1

Done as suggested. We reduced the scale to 0.9 – 1.1

Fig. 5 upper panels. X-axis caption should be T and not T'

Thanks, this has been corrected.

Fig. 6. The black-shaded land areas whenever Ep values are too low are somewhat confusing.

We actually found this presentation less confusing than showing the continental contours by mere black lines. Since this appears to be a matter of taste we have left this figure as it was.